

# PHACTR1 is associated with disease progression in Chinese Moyamoya disease

Yongbo Yang[1], Jian Wang[2], Qun Liang[3], Yi Wang[1], Xinhua Chen[1], Qingrong Zhang[1], Shijie Na[1], Yi Liu[4], Ting Yan[5], Chunhua Hang[1] and Yichao Zhu[6,7]

[1] Department of Neurosurgery, The Affiliated Drum Tower Hospital of Nanjing University Medical School, Nanjing, Jiangsu, China
[2] Department of Neurosurgery, The Affiliated Changzhou No. 2 People's Hospital of Nanjing Medical University, Changzhou, Jiangsu, China
[3] Drum Tower Clinical Medical College, Nanjing Medical University, Nanjing, Jiangsu, China
[4] Department of Neurosurgery, West China Hospital, Sichuan University, Chengdu, Sichuan, China
[5] Safety Assessment and Research Center for Drug, Pesticide and Veterinary Drug of Jiangsu Province, Nanjing Medical University, Nanjing, Jiangsu, China
[6] Department of Physiology, Nanjing Medical University, Nanjing, Jiangsu, China
[7] State Key Laboratory of Reproductive Medicine, Nanjing Medical University, Nanjing, Jiangsu, China

Corresponding authors
Chunhua Hang,
hang_neurosurgery@163.com
Yichao Zhu, zhuyichao@njmu.edu.cn

## ABSTRACT

Moyamoya disease (MMD) is a progressive stenosis at the terminal portion of internal carotid artery and frequently occurs in East Asian countries. The etiology of MMD is still largely unknown. We performed a case-control design with whole-exome sequencing analysis on 31 sporadic MMD patients and 10 normal controls with matched age and gender. Patients clinically diagnosed with MMD was determined by digital subtraction angiography (DSA). Twelve predisposing mutations on seven genes associated with the sporadic MMD patients of Chinese ancestry (*CCER2, HLA-DRB1, NSD-1, PDGFRB, PHACTR1, POGLUT1*, and *RNF213)* were identified, of which eight single nucleotide variants (SNVs) were deleterious with CADD PHRED scaled score > 15. Sanger sequencing of nine cases with disease progression and 22 stable MMD cases validated that SNV (c.13185159G>T, p.V265L) on *PHACTR1* was highly associated with the disease progression of MMD. Finally, we knocked down the expression of PHACTR1 by transfection with siRNA and measured the cell survival of human coronary artery endothelial cell (HCAEC) cells. PHACTR1 silence reduced the cell survival of HCAEC cells under serum starvation cultural condition. Together, these data identify novel predisposing mutations associated with MMD and reveal a requirement for PHACTR1 in mediating cell survival of endothelial cells.

## INTRODUCTION

Moyamoya disease (MMD) is a progressive stenosis at the terminal portion of internal carotid artery (ICA) with compensatory development of a hazy network of basal collaterals called Moyamoya vessels (*Scott & Smith, 2009*; *Suzuki & Takaku, 1969*). The prevalence of MMD is the highest in East Asian countries, including Japan, Korea and China (*Kuroda &*

*Houkin, 2008*; *Miao et al., 2010*). The annual incidences of MMD in China and Japan are 0.43 and 0.54 per 100,000, which are significantly higher than that in the USA (0.086 per 100,000) and Europe (0.3 per 100,000) (*Kraemer, Heienbrok & Berlit, 2008*; *Kuriyama et al., 2008*; *Miao et al., 2010*; *Uchino et al., 2005*). Epidemiology studies had revealed several risk factors associated with MMD, such as Asian ethnicity, female gender and family history (*Ganesan & Smith, 2015*). Although the heritability of MMD is unknown, the genetic components may play an important role in the etiology of MMD (*Inayama et al., 2018*).

Previous studies have explored and revealed several genetic loci associated with MMD, such as 3q24-p26, 6q25, 8q23, 17q25 (*Ikeda et al., 1999*; *Inoue et al., 2000*; *Sakurai et al., 2004*; *Yamauchi et al., 2000*). A polymorphism of c.14576G>A in the *RNF213* gene (*RNF213*) on the 17q25-ter region was identified as a novel susceptibility gene for MMD in Japanese and Chinese populations with a founder effect (*Liu et al., 2011*). *RNF213* is correlated with early onset and severe forms of MMD. Recently, rare variants on the C-terminal of *RNF213* were found correlated with MMD arteriopathy in patients of European ancestry (*Guey et al., 2017*), while a genome-wide association study involving a large case-control study among Chinese ancestry revealed 10 novel loci could be responsible for MMD, which extended our knowledge of MMD (*Duan et al., 2018*). However, the etiology of MMD is far more complicated than we expected, and further study on genome wide association is necessary.

Disease progression is among the most frequent reason for clinical symptomatic events. The clinical characteristics has been increasingly recognized as a prevalent cause of disease progression in those patients with natural course, such as early age onset, autoimmune factors and family heritability, but the evaluation is mostly focused on clinical characteristics (*Dlamini, Muthusami & Amlie-Lefond, 2019*; *Grangeon et al., 2019*; *Jiang et al., 2018*; *Kim & Jeon, 2014*). The relationship between genetic risk factor and disease progression in MMD patients remains unknown.

In this study, we performed a case-control design with whole-exome sequencing analysis on 31 sporadic MMD patients (nine cases with disease progression and 22 stable MMD cases) and 10 normal controls with matched age and gender. Predisposing mutations were discovered and then validated by Sanger sequencing. We identified 12 predisposing mutations on seven genes associated with the sporadic MMD patients of Chinese ancestry and further validated that SNV (c.13185159G>T, p.V265L) on *PHACTR1* was highly associated with the disease progression of MMD. Finally, we evaluated the effect of PHACTR1 on cell survival of endothelial cells.

## MATERIAL AND METHODS

### Subjects

We enrolled 31 MMD subjects and 10 non-related healthy controls with matched age and gender during 2018 to 2019 at Department of Neurosurgery, The Affiliated Drum Tower Hospital of Nanjing University Medical School, Nanjing, China. Subjects recruited into this study were all Chinese ancestry. The study was reviewed and approved by the research ethics committee, Nanjing Drum Tower Hospital of Nanjing University Medical School (2018-173-01). Written informed consent was taken from each participant.

Patients with MMD determined by digital subtraction angiography (DSA) were based on the guidelines established by the Japanese Research Committee on Moyamoya disease of the Ministry of Health, Welfare and Labor, Japan (RCMJ) (*Fukui, 1997*). Information on family histories, gender, age, onset symptoms were obtained by interview. Cases with additional evidence of arthrosclerosis, meningitis, autoimmune diseases, brain neoplasm, Down syndrome, Recklinghausen disease, irradiation or other obvious specific etiologies were excluded. All included MMD subjects all first time came to the hospital and diagnosed with the MMD. All these MMD subjects did not have any drug treatment for MMD or other conditions.

### Treatment and clinical follow-up

Through the use of single-photon emission computed tomography (SPECT), cerebral blood flow (CBF) was semi-quantitatively measured for all MMD patients 1 month after the initial onset. After preoperative imaging evaluation, surgical revascularization, including superficial temporal artery to middle cerebral artery anastomosis combined with encephalo-duroarterio-synangiosis (EDAS), was conducted in the hemisphere with intracranial hemorrhage or with more severe ischemia. If a patient elected not to undergo surgery, conservative management was used instead. After baseline investigation and surgical intervention, all patients underwent clinical follow-up. Brain MRI and MRA were performed every 6 to 12 months. During the follow-up period, if major intracranial artery stenosis progression was suspected on MRA or cerebrovascular events occurred, cerebrovascular DSA was performed within 1 week. The presence or absence of disease progression was evaluated at the final follow-up visit which defined as the progression of any major intracranial artery stenosis >50% in the ICA and/or the posterior cerebral artery (PCA). Representative cases with disease progression are shown in Fig. 1.

### Blood sample extraction and storage

Three tubes of 15 mL peripheral blood samples were collected and subjected to DNA extraction (Qiagen). DNA extraction method is seen in the exome sequencing analysis section. DNA samples were used immediately to next experiments, or store at −80 °C for later usage.

### Whole-exome sequencing and Sanger sequencing

We conducted exome sequencing analysis on 31 sporadic MMD subjects and 10 non-related healthy control subjects. Peripheral blood samples were collected from the subjects, and DNA was extracted using the QIAamp$^{TM}$ DNA and Blood Mini kit (Qiagen$^{TM}$, Munich, Germany), and sheared using acoustic fragmentation (Covaris) and purified using a QIAquick PCR Purification Kit (Qiagen). The whole-exome DNA library was sequenced on an Illumina HiSeq X Ten platform and performed as previously described (*Stachler et al., 2015*).

The Sanger sequencing was performed in Genscript Ltd. (Nanjing, China).

### Bioinformatics analysis

Raw sequencing reads and all qualified reads were processed with an in-house bioinformatics pipeline, which followed as previously described (*Stachler et al., 2015*).

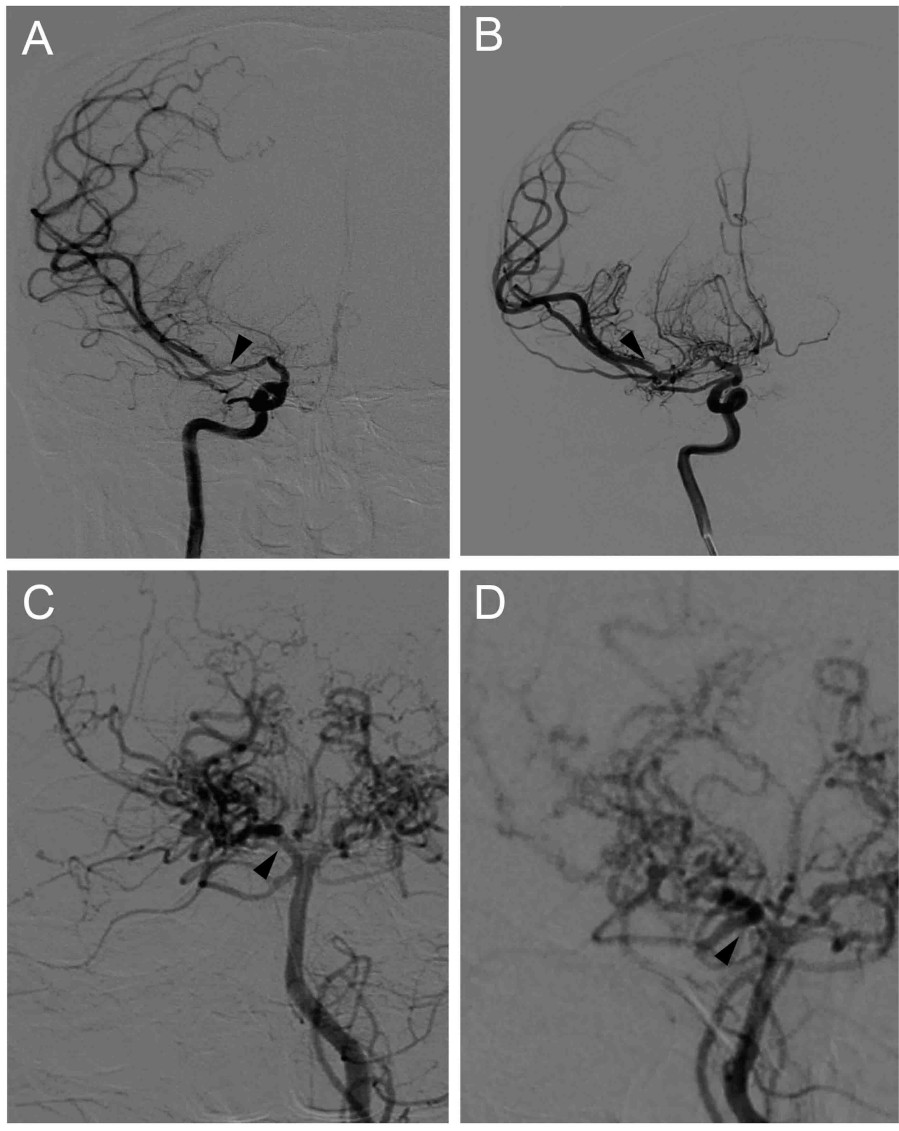

**Figure 1  Representative cases of disease progression in MMD.** (A) Right internal carotid angiograms of a 48 y/r man, showing artery stenosis progression in the proximal portion of right MCA (arrows). (B) Right posterior cerebral angiograms of a 51 y/r woman, showing artery stenosis progression in the proximal portion of right PCA (arrows).

Duplicated fragments were marked by Picard v1.141. After converting the data into bam format, GATK BaseRecalibration module was used to improve the base quality and then HaplotypeCaller module was used to discover genetic variants (SNV/INDEL). SnpEff 4.3i and Gemini v0.18.0 were used for functional annotation with Online Mendelian Inheritance in Man (OMIM), the Exome Aggregation Consortium (ExAC) Browser, MutationTaster2 and the Combined Annotation Dependent Depletion (CADD).

## Variants filtering and selection

Variants were excluded if they had a call rate <98%, major allele frequency <1%, abnormal heterozygosity >(mean ± 3SD), or significant deviation from Hardy–Weinberg equilibrium among controls ($P_{hwe} < 1 \times 10^{-4}$). The non-coding, synonymous, impact_severity=low, MAF (minor allele frequency) >0.01 variants were excluded from the raw data. SO_IMPACT=HIGH was referenced using MAF database including ESP, 1KG, ExAC database. The definition of "SO_IMPACT=HIGH" is based on the Gemini (https://gemini.readthedocs.io/en/latest/content/database_schema.html#details-of-the-impact-and-impact-severity-columns). The filtered variants was compared with known MMD related genes, such as *RNF213, CCER2, HLA-DRB1*. SNVs with CADD PHRED scaled score >15 were regarded as deleterious.

## siRNA transfection and Western blotting

Human coronary artery endothelial cell (HCAEC) was kindly gifted from Dr. Shengnan Li (Nanjing Medical University) and cultured in DMEM plus 10% fetal bovine serum (FBS, Hyclone, Waltham, MA). SiRNA targeting *PHACTR1* (sc-95456, Santa Cruz Biotechnology, Santa Cruz, CA) and scrambled siRNA were transiently transfected into HCAECs using Lipofectamine 2000 Reagent (Invitrogen, Carlsbad, CA). HCAEC cells 48 h after transfection were lysed with ice-cold RIPA lysis buffer. The SDS-PAGE was performed as the standard procedure. Anti-PHACTR1 (sc-514800, Santa Cruz Biotechnology) and anti- β-actin (Sigma, St. Louis, MO) were used. Protein bands were visualized with ECL reagent (Thermo Scientific, Rockford, IL) and recorded by Tanon 5200 Multi Imaging Workstation (Tanon, Shanghai, China).

## Cell survival assay

3-(4,5-dimethylthiazol-2-yl)-2,5-diphenyltetrazolium bromide (MTT) was used for the assessment of HCAEC cell survival and performed as previously described (*Zhu et al., 2012*).

## Statistical analysis

Continuous variables were described as mean ± SD, and categorical variables were presented as number and percentage. An independent $t$-test, and a chi-square or Fisher exact test were used to compare patients with and without disease progression. A $P$ value <0.05 was considered significant. All statistical analyses were performed using SPSS 23.0 (IBM, Chicago, Illinois).

# RESULTS

## Characteristic of subjects and variants quantity

The mean age (years ± SD) of all enrolled cases was 33 ± 8.46, and sex ratio of male: female was 0.34. The mean age and sex ratio of these healthy controls were 35.1 ± 6.58 and 0.3, which matched the disease group (Table 1). Of the 31 patients, 14 (45.2%) presented as intracranial hemorrhage and 17 (54.8%) presented as ischemic stroke at baseline. Among these 31 subjects, 9 of them suffered from disease progression in 10 hemispheres during the

**Table 1  Demographic and clinical data of MMD cases and healthy controls.**

| Characteristics | MMD group | Healthy group |
|---|---|---|
| Age (Mean ± SD) | 33 ± 8.46 | 35.10 ± 6.58 |
| Gender (Male/Female) | 0.34 | 0.30 |
| Ages < 18y (%) | 7 (22.58%) | 2 (20%) |
| Ages ≥ 18y (%) | 24 (77.42%) | 8 (80%) |
| Family history | No | No |
| Disease progression (%) | 9 (29.03%) | — |
| Stable disease (%) | 22 (70.97%) | — |
| Other combined severe conditions | No | No |

follow-up periods including 8 in the anterior circulation and 2 in the posterior circulation, and the other 22 cases were diagnosed as stable MMD disease.

By comparing with whole-exome sequencing results of the 31 sporadic MMD subjects with the human genome GRCh37.75, total 15,206 genes were successfully picked out. The main sequencing quality and depth were 85.42 M ±11.32 M reads, 99.69 ± 0.16 map (%), 111.87 ± 12.42 depth, suggesting a high quality of whole-exome sequencing. Total 196,365 variants were found including 176,582 site nucleotide polymorphisms (SNP), 8,906 insertions (INS) and 10,877 deletions (DEL) (Table 2). Total 117,748 (51.359%) variants were mis-senses, 1,525 (0.665%) variants were non-senses, and 109,992 (47.976%) variants were silent types. By comparing with the healthy subjects, 10,241 variants in exons were discovered.

## Significant predisposing mutations associated with disease progression

By comparing with the known MMD-related genes, 12 predisposing mutations on seven gene exons had from medium to high impact on gene modification (Table 3). These genes includes *RNF213, CCER2, HLA-DRB1, NSD-1, PDGFRB, PHACTR1* and *POGLUT1,* 8 of these SNVs were deleterious with CADD PHRED scaled score >15. Next, we performed Sanger sequencing to validate the association between predisposing mutations and MMD disease progression. SNV (c.13185159G>T, p.V265L) on *PHACTR1* was found in 3 cases with disease progression and shown significantly associated with its progression.

## The effect of PHACTR1 on cell survival

We hypothesized that PHACTR1, one of seven genes mentioned above, may mediate cell survival of human endothelial cells. We knocked down the expression of PHACTR1 by transfection with siRNA targeting PHACTR1 (Fig. 2A), and then measured the cell survival of HCAEC cells (Fig. 2B). Western blotting revealed that siRNA targeting PHACTR1 largely downregulated the expression of PHACTR1 in HCAEC cells, insulting in a ∼80% decrease (Fig. 2A). After the successive 96 h measurement of cell survival rate, we found that PHACTR1 silence reduced the cell survival of HCAEC cells under serum starvation cultural condition for 96 h, but not for 24∼72 h (Fig. 2B).

**Table 2  Total amount of variants found on every chromosome.**

| Chromosome | Length | Variants |
|---|---|---|
| 1 | 249,250,621 | 19,250 |
| 2 | 243,199,373 | 13,079 |
| 3 | 198,022,430 | 10,188 |
| 4 | 191,154,276 | 7,014 |
| 5 | 180,915,260 | 8,074 |
| 6 | 171,115,067 | 12,066 |
| 7 | 159,138,663 | 10,358 |
| 8 | 146,364,022 | 6,694 |
| 9 | 141,213,431 | 8,301 |
| 10 | 135,534,747 | 8,454 |
| 11 | 135,006,516 | 11,547 |
| 12 | 133,851,895 | 9,960 |
| 13 | 115,169,878 | 3,211 |
| 14 | 107,349,540 | 6,748 |
| 15 | 102,531,392 | 7,578 |
| 16 | 90,354,753 | 9,221 |
| 17 | 81,195,210 | 10,799 |
| 18 | 78,077,248 | 3,263 |
| 19 | 59,128,983 | 13,925 |
| 20 | 63,025,520 | 4,873 |
| 21 | 48,129,895 | 2,394 |
| 22 | 51,304,566 | 5,289 |
| X | 155,270,560 | 3,838 |
| Y | 59,373,566 | 241 |
| Total | **3,095,677,412** | **196,365** |

## DISCUSSION

All of the recruited sporadic MMD patients are all first-time being diagnosed with this disease, and did not receive any clinical treatment or drugs administration for any other medical conditions. *RNF213* was identified as a novel susceptible gene for MMD in East Asian population. This study identified 4 novel susceptibility loci and confirmed the previous reported susceptibility gene *RNF213*. Furthermore, we identified *HLA-DRB1* variants are found in the sporadic MMD patients. *HLA-DRB1* play a central role in the immune system by presenting peptides derived from extracellular proteins (*Ji et al., 2018*; *Lauterbach et al., 2014*). Although MMD patients with auto-immune diseases were excluded from this study, immune system dysfunction is still associated with the MMD pathophysiological process. *CCER2* was reported as a biomarker for MMD by Japanese colleagues (*Mukawa et al., 2017*). *NSD-1* is a transcriptional factor (*Jo et al., 2016*). In this study, we found a variant on *CCER2* locus and two mutations on *NSD-1*.

Our results suggest that cytoskeleton system and cardiovascular development are involved with the MMD pathological process. *PDGFRB* and *PHACTR1* work on the actin cytoskeleton system, and *PDGFRB* regulates cardiovascular development

Yang et al. (2020), *PeerJ*, DOI 10.7717/peerj.8841

Peerj

**Table 3 Associations between predisposing mutations and disease progression in MMD.**

| Locus | Gene | Position | Ref | Alt[a] | Codon change | Amino acid change | Qual | Depth | Impact | CADD raw score | CADD PHRED scaled score[d] | Progression/ Stable | wild type[b] | Mutant[b] | P value[c] |
|---|---|---|---|---|---|---|---|---|---|---|---|---|---|---|---|
| 19q13.2 | *CCER2* | 39401715 | C | T | c. 39401715C>T | p.G67R | 1763.4 | 3451 | Medium | 1.187 | 14.37 | Progression | 9 | 0 | 0.71 |
| | | | | | | | | | | | | Stable | 21 | 1 | |
| 6p21.32 | *HLA-DRB1* | 32549352 | G | C | c. 32549352G>C | p.P212A | 372.77 | 6450 | Medium | 1.940 | 18.63 | Progression | 8 | 1 | 0.71 |
| | | | | | | | | | | | | Stable | 22 | 0 | |
| | | 32551942 | T | C | c. 32551942T>C | p.D105G | 2688.4 | 16207 | Medium | 2.289 | 21.90 | Progression | 8 | 1 | 0.71 |
| | | | | | | | | | | | | Stable | 22 | 0 | |
| 5q35.3 | *NSD1* | 176562643 | T | C | c. 176562643T>C | p.I180T | 1044.4 | 2736 | Medium | 2.862 | 23.30 | Progression | 9 | 0 | 0.71 |
| | | | | | | | | | | | | Stable | 21 | 1 | |
| | | 176638711 | A | G | c. 176638711A>G | p.H1104R | 1108.4 | 2199 | Medium | 0.003 | 2.68 | Progression | 9 | 0 | 0.71 |
| | | | | | | | | | | | | Stable | 21 | 1 | |
| 5q32 | *PDGFRB* | 149513304 | G | A | c. 149513304G>A | p.P260L | 2883.4 | 4330 | Medium | 2.866 | 23.30 | Progression | 9 | 0 | 0.71 |
| | | | | | | | | | | | | Stable | 21 | 1 | |
| 6p24.1 | *PHACTR1* | 13185159 | G | T | c. 13185159G>T | p.V265L | 4277.88 | 4183 | Medium | 2.622 | 22.80 | Progression | 6 | 3 | *0.019* |
| | | | | | | | | | | | | Stable | 22 | 0 | |
| 3q13.33 | *POGLUT1* | 119211265 | A | T | c. 119211265A>T | p.M387L | 1367.4 | 2013 | Medium | 0.846 | 12.20 | Progression | 9 | 0 | 0.71 |
| | | | | | | | | | | | | Stable | 21 | 1 | |
| | | 78291014 | A | T | c. 78291014A>T | p.Q995H | 736.4 | 3008 | Medium | 0.677 | 10.91 | Progression | 9 | 0 | 0.71 |
| | | | | | | | | | | | | Stable | 21 | 1 | |
| | | 78319385 | T | G | c. 78319385T>G | p.I2466S | 3066.4 | 3554 | Medium | 3.499 | 25.10 | Progression | 9 | 0 | 0.71 |
| | | | | | | | | | | | | Stable | 21 | 1 | |
| 17q25.3 | *RNF213* | 78320960 | T | C | c. 78320960T>C | p.V2991A | 2086.4 | 3562 | Medium | 1.924 | 18.48 | Progression | 9 | 0 | 0.71 |
| | | | | | | | | | | | | Stable | 21 | 1 | |
| | | 78321631 | A | G | c. 78321631A>G | p.I3215V | 1814.4 | 3802 | Medium | 2.739 | 23.10 | Progression | 9 | 0 | 0.71 |
| | | | | | | | | | | | | Stable | 21 | 1 | |

**Notes.**
[a] Missence variant.
[b] According to Sanger sequencing.
[c] *P* value for χ2 test.
[d] CADD PHRED-like scaled C-scores = -10*log_10 (rank/total), the recommended deleterious threshold was >15 for scaled C-scores.

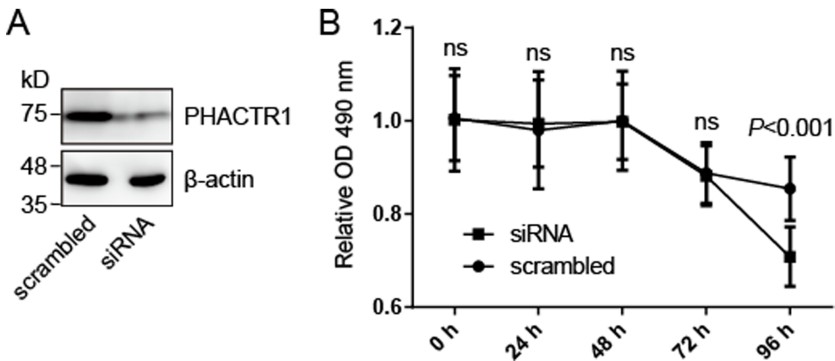

**Figure 2  PHACTR1 silence reduces the cell survival of HCAEC cells.** (A) HCAEC cells were transiently transfected with siRNA targeting PHACTR1 or scrambled siRNA. Western blotting verified the knock-down efficiency of siRNA targeting PHACTR1. β-actin as the loading control. (B) HCAEC cells transfected with siRNA targeting PHACTR1 or scrambled siRNA were cultured under serum starvation cultural condition for successive 96 h. Cell survival rate was assessed by cell survival assays. PHACTR1 silence reduced the cell survival of HCAEC cells under serum starvation cultural condition for 96 h.

(*Onel et al., 2018*; *Perez-Hernandez et al., 2016*). *POGLUT1* works on the NOTCH signaling pathway to regulate developments (*Wu et al., 2017*). According to Sanger sequencing, we found that SNV (c.13185159G>T, p.V265L) on *PHACTR1* was significantly associated with MMD disease progression. Owing to the lack of appropriate animal model for MMD, we provisionally checked the effect of genes associated with MMD susceptibility on immortal endothelial cells. We found PHACTR1 mediates the cell survival of endothelial cells under serum starvation cultural condition. Nevertheless, the precise mechanisms of MMD needs to be further studied on molecular and cellular levels.

The main limitation of the study is the small size of samples. The subjects are difficult to recruit, we have attempted to perform a meta-analysis to strengthen or our conclusions. We have gone through three databases and obtained 76 articles about MMD (Pubmed, $n = 23$; Embase, $n = 33$; Web of science, $n = 20$). The duplicated articles ( $n = 42$) were removed. Then, 29 articles were excluded through sceening title or/and abstract, full text, due to review, animal studies, case report and/or unrelated outcomes, moyamoya syndrome or other diseases. After reading the full text of 13 sceened articles, we have not found these 8 SNVs mentioned in Table 3. So, we are unfortunately unable to perform a meta-analysis in the present research status.

## CONCLUSIONS

We perform whole-exome sequencing on 31 sporadic MMD subjects and 10 healthy volunteers, and identify 12 predisposing mutations of seven genes (*RNF213, CCER2, HLA-DRB1, NSD-1, PDGFRB, PHACTR1* and *POGLUT1*) associated with MMD and SNV (c.13185159G>T, p.V265L) on *PHACTR1* associated with its progression. We preliminarily provide the rational evidence of the effect of PHACTR1 on endothelial cell survival and indicate its involvement in the pathophysiological process of MMD.

**Abbreviations**

| | |
|---|---|
| **MMD** | Moyamoya disease |
| **HCAEC** | human coronary artery endothelial cell |
| **DSA** | digital subtraction angiography |
| **MCA** | middle cerebral artery |
| **SNV** | single nucleotide variant |
| **MAF** | minor allele frequency |

### Funding

This work was supported by grants from the Key Project supported by Medical Science & Technology Development Foundation of Nanjing Department of Health (ZKX15014) to Yongbo Yang, the Jiangsu Postdoctoral Foundation of Jiangsu Provincial Department of Human Resources & Social Security (1501122B) to Yongbo Yang, and the Science and Technology Foundation of Nanjing Medical University (2017NJMU001) to Ting Yan. The funders had no role in study design, data collection and analysis, decision to publish, or preparation of the manuscript.

### Grant Disclosures

The following grant information was disclosed by the authors:
Medical Science & Technology Development Foundation of Nanjing Department of Health: ZKX15014.
Jiangsu Postdoctoral Foundation of Jiangsu Provincial Department of Human Resources & Social Security: 1501122B.
The Science and Technology Foundation of Nanjing Medical University: 2017NJMU001.

### Competing Interests

The authors declare there are no competing interests.

### Author Contributions

- Yongbo Yang conceived and designed the experiments, performed the experiments, analyzed the data, prepared figures and/or tables, and approved the final draft.
- Jian Wang conceived and designed the experiments, analyzed the data, prepared figures and/or tables, and approved the final draft.
- Qun Liang performed the experiments, prepared figures and/or tables, and approved the final draft.
- Yi Wang analyzed the data, prepared figures and/or tables, and approved the final draft.
- Xinhua Chen, Qingrong Zhang, Shijie Na and Yi Liu performed the experiments, authored or reviewed drafts of the paper, and approved the final draft.
- Ting Yan performed the experiments, analyzed the data, authored or reviewed drafts of the paper, and approved the final draft.
- Chunhua Hang conceived and designed the experiments, authored or reviewed drafts of the paper, and approved the final draft.

- Yichao Zhu conceived and designed the experiments, analyzed the data, prepared figures and/or tables, authored or reviewed drafts of the paper, and approved the final draft.

## Human Ethics

The following information was supplied relating to ethical approvals (i.e., approving body and any reference numbers):

The study was reviewed and approved by the research ethics committee, Nanjing Drum Tower Hospital of Nanjing University Medical School (2018-173-01).

## Data Availability

The raw measurements available in the Supplemental Files.

## Supplemental Information

Supplemental information for this article can be found online at http://dx.doi.org/10.7717/peerj.8841#supplemental-information.

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
