# Peer review of "PHACTR1 is associated with disease progression in Chinese Moyamoya disease"

_PeerJ, doi:10.7717/peerj.8841_

## Round 0.1 · original submission · Major Revisions

Our reviewers obviously agree that this is a very important question, and the major source of skepticism is "effect size." The small number of subjects who showed the predicted effect is problematic, as you know. The best answer is to increase sample size.

The paper does not report several points, as our reviewers said. These omissions should be corrected.

If subjects are difficult to recruit, a meta-analysis of several papers MIGHT be considered to strengthen or weaken the conclusions. A good statistician should be consulted.

As you know, there are now many public archives for diagnostic conditions, and that would be even better, but I don't know it if would be practical.

All the reviewers' comments should be addressed in your rebuttal letter for any revision

·

Basic reporting

no comment

Experimental design

no comment

Validity of the findings

no comment

Additional comments

In this study, yongbo Yang et al try to find potential association between gene mutations and disease progression in Chinese MMD. The results from whole-exome sequencing analysis on 31 sporadic MMD patients revealed that PHACTR1was associated with disease progression. However,of these patients, only 9 patients had disease progression and 3 of them had PHACTR1 mutant. Though statistical analysis was positive, I still query the result reliability. More sample would be better.
1 All patients were ischemic type?
2 What is the criterion for disease progression ? Figure 1 only provide the DSA at admission, more details of follow up results of DSA should be added(disease progression evidence)
3 More limitations should added!

Reviewer 2 ·

Basic reporting

1. In the discussion, the authors described the known relationship between RNF213 and MMD in detail. However, it is not the major finding in the manuscript. The author should consider rewriting this part. In addition, the limitation of the study was not mentioned in the discussion.
2. In the abstract, it is stated that “8 single nucleotide variant (SNV) were deleterious with PHRED scaled score >15”, which is not accurate and misleading. It should be PHRED-like scaled C-scores, while PHRED scaled score was considered as the quality measure of the sequencing results.
3. Some grammar mistakes should be corrected, such as line 133. “was performing” should be changed to “was performed”.
4. Since no RNA was used in the study, why the RNAeasy kit was mentioned in the line 118.
5. line 166, it should be “Santa Cruz” instead of “anta Cruz”
6. Some abbreviations were not explained. Such as line 152, MAF. I assume it should be “minor allele fraction”? if so, it should be “MAF <0.01” as the threshold. Also in line 153, it is not metioned the term “SO_IMPACT=HIGH”.

Experimental design

1. The author used Chi-square test to determine the associations between SNV and disease progression. Considering the small sample size and some expected numbers are less than 5, it does not meet the prerequisite of Chi-square. An exact test should be performed.
2. Although the transfection of siRNA showed the reduction of cell survival after 96 h, it will be more persuasive if the rescue experiment is performed.

Validity of the findings

1. In the results, the sequencing quality and depth was not mentioned, which is a very important factor to evaluate the results.
2. It is mentioned in the methods that “The filtered variants was compared with known MMD 155 related genes, such as RNF213, ACTA2, BRCC3, GUCY1A3, CCER2, R179H, C536G>R, HLA-156 B51, HLA-B35, HLA-B67, HLA-DR1, HLA-DQB1*0502”. Only some of the genes were mentioned in the following text. How many of the known related genes were identified in this work, as well as the potential explanation was not discussed.

Additional comments

The manuscript of Yang Y et al. performed whole-exome sequencing analysis between 31 patients and 10 healthy controls to identify the possible variants related to the progression of Moyamoya disease. They have found twelve predisposing mutations from 7 genes which may be potential, and 8 of them were predicted as deleterious by the bioinformatics tools. Moreover, they have performed experiments to confirm one of the SNVs in PHACTR1 and its function. Generally, the methods and results are clear; however, several major concerns should be addressed before this can be considered further publication.

---

## Round 0.2 · accepted · Accept

I am happy that this paper is suitable for publication. It is important to work on these rare phenotypes and get the data out to stimulate collaboration and additional research.

Reviewer 2 ·

Basic reporting

Now the manuscript is in a good form to be published.

Experimental design

The issues have been addressed.

Validity of the findings

The findings are substantially improved.

Additional comments

Manuscript of "PHACTR1 is associated with disease progression in Chinese Moyamoya disease" has been improved substantially over an earlier version. There are no further comments on the paper.